# High Impostors Are More Hesitant to Ask for Help

**DOI:** 10.3390/bs14090810

**Published:** 2024-09-12

**Authors:** Si Chen, Lisa K. Son

**Affiliations:** 1Department of Psychology, The University of Hong Kong, Hong Kong, China; 2Department of Psychology, Barnard College, Columbia University, New York, NY 10027, USA; lson@barnard.edu

**Keywords:** help-seeking, impostorism, metacognition, metacognitive control, math learning

## Abstract

Help-seeking behavior requires both components of metacognition—monitoring (being aware of the need for help) and control (initiating the help-seeking action). Difficulties in initiating help-seeking, therefore, can be indicative of a metacognitive breakdown, for instance, when a student believes that a gap in knowledge is something to hide. To explore the relationship between knowing that one needs help and actually seeking it, we examined the potential influences of impostorism, which refers to the feeling of being a “fraud”, despite one’s objective accomplishments. Participants were asked to solve math reasoning and verbal reasoning insight problems, while also being given a “help” button that could be pressed at any time in order to get the solution. Results showed that, overall, students were more likely to ask for help with math than verbal reasoning problems—help also correlated with boosted performance. There was also a slight indication that individuals who scored relatively high on impostorism were numerically less likely to seek help and waited longer to do so for the math problems. Our findings suggest that a fear of being exposed as an impostor may hinder one’s help-seeking behaviors, especially in more challenging subjects, such as math.

## 1. Introduction

For decades now, there has been a growing interest in metacognition, or our ability to self-reflect and self-regulate. In the classic framework, metacognition comprises two separate but related processes: (1) monitoring and (2) control. Early metacognitive studies were conducted to measure the accuracy of our monitoring judgments by looking for a positive correlation between our performance and our judgments of our knowledge [1,2,3,4]. Meanwhile, an important interest in the control process emerged almost in tandem [5,6], persisting with ongoing investigations [7]. The question of “Why possess a monitoring skill?” was exciting, and the answer was that we needed to be able to accurately assess our knowledge so that we could select the optimal control strategies for filling gaps in our knowledge [8,9]. The current study examined our decisions for a particular strategy used to fill in those gaps: help-seeking behavior.

We were motivated by the fact that some students would hesitate, or outright refuse, to initiate help-seeking, despite their awareness of their need for help. One possible explanation, we proposed, for this gap between intact monitoring and failed control could lie in an individual’s experience with the impostor phenomenon. Previous studies have, in fact, suggested a link between impostor fear and decision-making in the field of STEM [10,11]. Following this line of research, the current study focused on the relationship between the impostor phenomenon and help-seeking behaviors, paying particular attention to the difference in math (STEM) and verbal (non-STEM) domains. Drawing on previous findings, we hypothesized that individuals who scored relatively high on the impostorism scale would be less likely to seek help compared to those who scored relatively low on the impostorism scale, and that this would be especially true for STEM learning, such as when solving math-related problems.

### 1.1. Metacognitive Control of Help Seeking

In almost all of the literature on metacognition, the control process has been examined under conditions where participants were learning individually [12,13]. For instance, the simplest behavior that has been used as a measure of control has been time allocation [14], or, more broadly, information-seeking behavior [15]. In this paradigm, the learner is asked to monitor their ongoing learning by making judgments about their current knowledge. In general, data have shown that, from the perspective of the discrepancy-reduction model [16], allocating more study time to the judged-unknown materials than to the judged-known materials makes sense, and is the strategy that learners tend to use [17,18]. In other words, one of the most basic control decisions that we make using our metacognitive skills is one where we control when and how much more information should be sought.

While the seeking of information through study-time allocation is important, much of our lives are spent learning in situations not in solitude, but in the presence of others. In the classroom, for instance, a primary means to seek more information would not necessarily be to allocate more self-study time, but rather to speak up and ask the teacher (or a peer) for more information. Indeed, we are well aware of our social context when making metacognitive decisions. Control processes have been found to be weakened by “divided attention” in a social environment [19], and to make matters complicated, an individual’s metacognitive processes could also involve the monitoring and control processes of the larger group [20].

In a social context, the link between one’s monitoring judgments and subsequent control decisions can become complicated. It is easy to imagine how information-seeking when others are present may disrupt one’s normally effective metacognitive control decisions. Any of us can remember a situation where we knew that we were confused by the teacher’s lecture (a scenario where our monitoring was intact—we knew that we did not know), but we kept silent about our confusion. Or, when in a study group preparing for an upcoming test, we hesitated asking our peers to explain a fundamental proof that felt impossible. In both scenarios, the most sensical thing to do would be to seek more information from the logical source—the teacher or our peers. And yet, the idea of interrupting a learning session to fill in the gap one has in their knowledge—an obvious control strategy—can be difficult. Or, at least, the relationship between monitoring and help-seeking is not particularly a strong one [21]. This is despite findings that show that receiving feedback is a strong predictor of later performance [22]. The fear that we would be posing a “stupid question” and thus being exposed as a “fraud” in front of our teachers and peers can be an example of how impostorism can hinder our metacognitive decisions in a social context.

Metacognition studies have supported a link between social cognition and individual metacognitive control, as discussed earlier. Nonetheless, little is known about the particular mechanisms behind individual metacognitive breakdown when information-seeking occurs in social contexts such as the two scenarios described above. To address this gap, the current study aimed to investigate this specific type of metacognitive breakdown observed in individual help-seeking behavior by focusing on the role of the impostor phenomenon, which is described in more detail below.

### 1.2. The Impostor Phenomenon

In the above example, there are a number of reasons why people may be hesitant to seek help from the teacher, or in front of others. One is that owning up to “not knowing” could be embarrassing. Moreover, allowing others to “find out” that you don’t understand the material could lead to false assumptions, such as the idea that you did not appropriately prepare for class or that you are “a lazy or academically entitled student”. Perhaps even on a deeper level, help-seeking could be an affirmation of something you fear most—that you are not cut out for this class. This feeling is related to the possibility that you do not legitimately belong at a particular level, or that you were put in the class by some sort of error or luck, all common feelings for people who experience what is known as the Impostor Phenomenon (IP). As a result, individuals who have symptoms of feeling like a “fraud” may try to avoid help-seeking behavior, as help-seeking might be a confirmation of their fears.

IP has been a topic of growing interest over the past few decades. It is important to note that the phenomenon has often been used synonymously with phrases such as Impostor Syndrome [23]. In fact, in the lay population, the term “Impostor Syndrome” appears to be the most familiar, as the Wikipedia page on the topic uses the term as its primary name (although the website definition does begin by stating that the term “Impostor phenomenon was introduced in 1978… by Clance and Imes”). Despite the indiscriminate use of the two terms, the current study uses the term impostor phenomenon, or impostorism, to highlight the social, cognitive, and affective experience of feeling like an impostor as opposed to impostor syndrome, which often entails clinical symptoms that are directed at an individual level [24].

Early research had claimed that IP would be observed primarily in successful women [25]. However, subsequent studies have offered broader results. King and Cooley (1995), for instance, found that women who were still only in college displayed high Impostor ratings [26]. Topping and Kimmel (1985) found that male faculty members had higher impostor ratings than female faculty members [27]. Moreover, evidence from a systematic review suggested no gender difference in the rates of IP [28]. On the other hand, recent large-scale meta-analytical findings revealed that, despite previously mixed findings, women overall scored more highly than men on impostorism, with this gender disparity being more prominent in Europe and North America [29]. Further IP research has also suggested different effects of IP on performance across genders [28]. Given that the current sample consisted mostly of women in higher education, it aims to contribute to this ongoing exploration of the gender dimension of IP.

For the most part, IP has been shown to affect people who are at varying stages of a successful career. For example, scholars [30], clinical managers [31], librarians [32], marketing managers [33], and physicians [34] have all reported that they experienced fears of being an impostor quite often. Interestingly, those who are just beginning on the path of a successful career seem to be more prone to IP. Younger librarians and those with less longevity experienced Impostor-like feelings at a higher rate than more veteran librarians [32]. In addition, even those whose careers are judged to be highly prestigious are affected by the phenomenon—of medical residents surveyed, approximately 30% of family medicine residents (*n* = 175) and 44% of internal medicine residents (*n* = 48) experienced feelings of being an impostor [34]. Thus, a common irony of IP seems to be that it is often hidden, masked by what might look like (the cusp of) a thriving career. In other words, “appearing successful” or “perfect” or “model” seems to be a key ingredient of being an impostor. The primary aim of the current study was to examine the effects of IP on help-seeking. We hypothesized that people who, comparatively, scored higher on the impostor scale would have more difficulty seeking help, namely because of this need to hide one’s imperfections or gaps in knowledge.

One of the earliest key findings of IP was that those who scored higher on Imes’ Attribution Scale tended to attribute their successes to external factors such as luck, rather than to internal factors such as ability [35,36]. Using her scale, Imes (1979) also found early on that, when surveying a group of university faculty, men obtained higher impostor scores than did women, and that for all participants, the score was negatively related to faculty rank, self-esteem, and attribution of success to effort [36]. First and foremost, individuals who score relatively high on the impostor scales tend to attribute their successes to external factors, when compared to those who score low on the scales [27,35,37,38]. On the other hand, as was shown recently in Vaughn (2020), Low Impostors tended to attribute their success to the self (e.g., one’s effort and ability) [39]. This feature of the High Impostor, i.e., the lack of attributing success to your own self, describes the core reason for the phenomenon, namely, feeling like a fraud. Attributing random external factors to success, in turn, seems to give rise to other characteristics, such as the worry of being “found out” for one’s own illegitimacy [40]. It is not surprising, therefore, that Clance (1985) also found that High Impostors tended to discount positive feedback and disliked receiving praise or attention for their achievements [25].

Ample attribution and motivation evidence support the detrimental effects of ego involvement on students’ help-seeking. That is, the more a student believes that an academic failure is a reflection of one’s ability and structures their actions in terms of competitive rewards and social comparisons, the less likely it is that they will seek help [41,42]. However, little research has investigated how impostorism, which may be characterized by a high level of self-consciousness and ego involvement, also plays a part in influencing students’ help-seeking patterns.

Indeed, the STEM field has been heavily criticized for its pedagogical emphasis on ego-involved self-focus and fixed ability mindset beliefs [43,44]. Research on STEM impostor syndrome (STEM-IS) showed that STEM-IS often emerges in early adolescence when students first encounter STEM education environments in which they feel marginalized or do not belong due to the field’s particular pedagogical approaches [45]. Previous IP findings within higher STEM education highlight the negative impact of cultural mismatches on STEM pursuits, particularly for minority groups, including women, racial minorities, and first-generation college students [46,47]. Environmental factors such as racial discrimination, hypercompetitive learning environments, and group underrepresentation have all been identified as contributing factors to the individual experience of IP in the STEM field [47,48,49]. This domain-specific impostor experience was found to be negatively linked to one’s later development of STEM self-efficacy and to contribute to higher math anxiety and STEM attrition [10]. Past studies on math problem-solving showed that math anxiety could lead to metacognitive failures, such as metacognitive blindness, characterized by the lack of attention to problems, and metacognitive mirages, where students become overly focused on certain aspects of problem-solving, leading them to feel that something is always wrong [50]. These metacognitive failures can, in turn, disrupt students’ problem-solving performance in mathematics.

Thus, building on past IP and math anxiety findings, the current study aimed to examine the general relationship between impostorism and students’ likelihood of seeking help. Specifically, we were interested in the role of impostorism in relation to the experience of help-seeking in areas where anxiety is especially high, such as in the math domain, for underrepresented groups such as women. While we predicted that those who scored higher on the IP scale would seek help less often, we explored the possibility that this would be especially true in the domain of math.

### 1.3. The Current Study

We set out to understand a part of metacognitive control that has been virtually ignored—help-seeking in an academic or social setting, and the effects of impostorism. Through a review of the literature, we came to the following hypotheses: Based on previous data relevant to impostorism, we thought that the fear of being “found out”, i.e., revealing that one lacks certain knowledge, could be a significant basis for avoiding help-seeking—an important and appropriate control strategy. To test our hypothesis, we designed a study that would allow people to solve problems and ask for help, and to measure the likelihood of help-seeking and one’s reluctance to ask for help via response time. We also administered the Clance IP (CIP) Scale, as it had been a frequent measure of impostorism in the past, and predicted that those who scored relatively high on the scale—the High(er) Impostors—would be less likely than those who scored lower on the scale—the Low(er) Impostors—to ask for help, as the former would want to avoid revealing any lack of knowledge. More broadly, we wanted to highlight the potential obstacles to a critical metacognitive control strategy—asking for help—in an academic setting.

In addition to our main hypothesis, we compared help-seeking in two different academic domains: verbal reasoning (VR) and mathematical reasoning (MR). In past studies, the materials used to test impostorism and help-seeking behavior seemed to be domain-specific. For instance, Canning (2020) found that impostorism, which was also higher for first-generation students, was related to class engagement, attendance, dropout intentions, and grades for STEM fields [47]. In light of our growing understanding of math and STEM anxiety, we predicted that the High(er) Impostors would be especially reluctant to seek help for MR problems, as compared with VR problems, assuming that a lack of knowledge in the verbal realm may not be considered something “to be hidden”. However, we considered this comparison exploratory and let the numbers speak for themselves. To summarize, then, our hypotheses were as follows:

**Hypothesis** **1:**
*High(er) Impostors will be more likely than Low(er) Impostors to avoid help-seeking, particularly when solving MR problems.*


**Hypothesis** **2:**
*High(er) Impostors will be more reluctant (slower) than Low(er) Impostors to seek help, particularly when solving MR problems.*


## 2. Materials and Methods

All subjects gave their informed consent for inclusion before they participated in the study. The study was conducted in accordance with the Declaration of Helsinki, and the protocol was approved by the College’s Institutional Review Board (IRB—Project ID#2223-0530-023).

### 2.1. Participants

Based on an a priori power analysis (*α* = 0.05, *power* = 0.80), to reach an effect size of 0.25, we recruited 144 participants (98% women with an average age of 18 years old) from an all-women’s liberal arts college in the US, and who were enrolled in Introductory Psychology (both women and men could enroll as the courses are co-ed, but gender was heavily biased towards women). Participants were recruited through a Psychology Department website where students could sign up for course credit. We acknowledge that the limited range of diversity in the current sample may not generalize to the larger population but believe in the benefits of limiting the diversity at this point to see the clear effects of impostorism. In particular, given the makeup of the current sample, we also had the opportunity to think about symptoms related to math anxiety, which has been found to be higher in women than in men [51]. Thus, in the final section, we delve into a deeper discussion of help-seeking behavior as a function of domain—MR vs. VR—and what they suggest for the specific population of women at a prestigious school in the conclusion.

### 2.2. Materials

The problems consisted of 2 MR and 2 VR questions (see Appendix A). A series of problems were initially selected from the Internet and were presented to 4 separate college-aged “judges”. Based on the judges’ mean scores for difficulty level, the 4 problems rated as highest in difficulty were selected for this study. The purpose for choosing the most difficult problems was so that they would be more likely to require some help in order to solve them. That is, our goal was to have people know that they did not know, and, hopefully, seek help.

IP Measure: We present regression data using the raw IP scores (which could range from 20 to 100) resulting from the CIP scale. In addition, as a secondary perspective of the data and consistent with other IP studies, a between-subjects IP Group division was assessed [10,25,52]. The scale has been validated for both clinical and non-clinical populations in measuring individual traits and feelings, which include fears of evaluation, social expectation, and failure [53]. While scores on the CIP scale could range from 20 to 100, they could also be organized into 4 categories, described in Clance (1985) as follows: “If the total score is 40 or less, the respondent has few Impostor characteristics; if the score is between 41 and 60, the respondent has moderate IP experiences; a score between 61 and 80 means the respondent frequently has Impostor feelings; and a score higher than 80 means the respondent often has intense IP experiences” [25]. Thus, in general, a 60 could be thought of as the median score on the CIP scale. Note that for the current analysis below, we present findings where the median is shifted to slightly higher, at a median of 66, to make an analogous comparison with the current sample. “The higher the score, the more frequently and seriously the Impostor Phenomenon interferes in a person’s life” (p. 23) [25].

### 2.3. Design

As mentioned above, we provide two types of analyses: (1) a regression design comparing the CIP score with both Help-Seeking, or the frequency of asking for help, and Reluctance, or the time it took to ask for help, along with Domain (MR vs. VR), and (2) a group analysis where the main independent variables were IP group (Higher vs. Lower, median split, between-subjects), and Domain (MR vs. VR, within-subjects), resulting in a 2 × 2 design. Each participant was given 4 insight problems to solve, always in the same order: First, the 2 MR questions, followed by the 2 VR questions. The insight problems were chosen to assess the domain characteristics while controlling for potential variation in participants’ MR and VR skills. For instance, although the first MR question seems to test participants’ numerical reasoning skills (8 + 4 = 2; 7 + 3 = 0; 4 + 9 = 1; 8 + 5 = ?), the intended solution requires “thinking out of the box” and seeing that the shapes of the numbers matter (i.e., 8 means 2 because the number contains 2 circles). For each participant, a total accuracy score, out of the 2 questions for each domain, was calculated to get the mean scores across groups.

### 2.4. Procedure

When participants first entered the experiment room, they were asked to sign a consent form, wherein they were informed of the purpose of the study, namely to investigate patterns of problem-solving processes across math and verbal domains. They were also told of their rights to withdraw, of data confidentiality, and of the fact that minimal risk was associated with the study participation. Once participants indicated their consent to be involved in the study, they were directed to the experiment which was administered on Qualtrics via computers in individual lab rooms. They were provided with instructions, which informed them that they would be presented with 2 parts, each consisting of 2 different domain questions: MR problems followed by 2 VR problems. They were also told that if they had any trouble solving any of the problems, they would be able to get help by clicking a button on the screen which was labeled “Ask for Help”. By clicking the Help button, participants would be presented with instructions on how to solve the problem at the bottom of the page while being able to refer back to the problem listed at the top of the page (see Appendix A for the list of questions with instructions and solutions).

Each of the problems was presented one at a time, accompanied by the Help button. If at any time the Help button was pressed, the program recorded that “Help was sought”. In addition, the time duration between the question being displayed and the participants clicking the Help button was also recorded as a measure of Reluctance. All participants solved the MR problems followed by the VR problems.

After completing the 4 problems, the CIP scale (Clance, 1985) was administered to measure IP (as well as to be used to compare group IP data) [25]. At the end of the study, participants were debriefed on the true purpose of the study, which was to examine the relationship between IP and help-seeking behaviors across MR and VR domains. All participants were then given course credits for taking part in the study.

## 3. Results

Based on the scoring scheme for IP suggested by Clance (1985) [25], 2% of the participants fell below a score of 40, indicative of low impostorism feelings, 32% reported moderate impostorism (scores between 41 and 60), 50% reported frequent impostorism (between 61 and 80), and 16% reported intense impostorism (scores above 80) (see Figure 1 for the frequency distribution). The median IP score for all participants in the current study was 66 (range was 26 to 91). This basic shift in the IP data suggests that this particular sample was, on average, higher in levels of impostorism than other populations tested in the past, and could perhaps speak to a higher level of impostorism in a sample consisting largely of women in higher education, from where the construct IP was originally derived [30,39]. The implications of the demographic composition and IP distribution will be further elaborated in the discussion. To understand the impostorism effects in the current study, we first treated participants’ IP scores as a continuous variable and conducted regression analyses. However, as additional analyses, and to approach the shift in the IP data (from a median of 60 to a median of 66), we also present analyses using a split of the participants into two groups based on the median score for this specific sample. As indicated by the Shapiro–Wilk test for normality, we were able to use non-parametric tests to address the non-normality when calculating effects on the help-seeking frequency (W = 0.77, *p* < 0.001) and the time taken to seek help (W = 0.82, *p* < 0.001). We used correlations to understand the relationship between help-seeking and accuracy scores.

In the following sub-sections, we first present findings on the domain differences in participants’ accuracy scores and help-seeking frequencies in general. Then, we examine another aspect of help-seeking, namely how reluctant participants were to seek help, in addition to whether they sought help or not. Finally, we demonstrate the merit of help-seeking in improving participants’ performance for the current study.

### 3.1. Are the Domains of MR and VR Different in Terms of Difficulty?

Overall, as is shown in Figure 2, participants successfully answered more of the VR problems (60.1%) than the MR problems (25.2%). A Wilcoxon signed rank test showed that the effect of Domain was significant [z = 7.151, *p* < 0.001, d = 0.40]: there was no effect of IP level or interaction between IP and domain accuracy.

These data suggest that, from the outset, there was a difference between the VR and MR problems, and, given this, in addition to the fact that the participants were primarily women, we felt somewhat more confident in our prediction that the data might exhibit different levels of help-seeking behaviors, and perhaps a nuanced interaction. Specifically, participants may be more likely to seek help on the MR problems than the VR problems, given that it is less likely that they could solve the former on their own. At the same time, however, the likelihood of help-seeking might also differ by level of impostorism. If our original hypothesis holds—that people are less likely to seek help if they are worried about being seen as not knowledgeable—then we might see the largest dip in help-seeking for high(er) impostors on MR problems. That is, we expected that there might be an interaction—IP may not make a difference on the VR problems, but for the MR problems, high(er) Impostors would be less likely to seek help than low(er) Impostors.

### 3.2. Are There Differences in Help-Seeking Behavior as a Function of IP and Domain?

The mean number of times that people sought help for MR problems and VR problems were 0.50 and 0.40, respectively. Figure 3a presents the distribution of participants’ IP scores at different help-seeking frequencies for the two domains separately with the MR domain (left panel) and the VR domain (right panel). The Wilcoxon signed rank test showed that there was a significant effect of domain on the probability of help-seeking [z = 3.241, *p* = 0.001, d = 0.27], indicating that, on the whole, participants sought help for MR problems significantly more than for VR problems. No statistically significant interaction effect was found between IP scores and domains. As an additional perspective, Figure 3b presents the same data grouped by the median split of IP scores (median IP = 66). Here again, although not statistically significant, there is a numerical pattern showing that High(er) Impostors sought less help on average (mean = 0.45, SD = 0.43, median = 0.50, IQR = 1.00) than Low(er) Impostors (mean = 0.54, SD = 0.42, median = 0.50, IQR = 1.00) for MR problems. For VR problems, High(er) Impostors sought help on average (mean = 0.40, SD = 0.42, median = 0.50, IQR = 1.00) similarly compared to Low(er) Impostors (mean = 0.40, SD = 0.38, median = 0.50, IQR = 0.50).

### 3.3. Who Is More Reluctant to Seek Help?

Another way to understand people’s help-seeking strategies is to investigate the amount of time it takes for people to seek help, if at all. In other words, when people monitor their knowledge and realize that they need help, there may be differences as to the criterion that exists for actually seeking that help. Overall, 36.4% of the participants did not seek help for MR problems, and 44.1% did not do so for VR problems. More importantly, those who did not seek help also failed to provide the correct answers in both MR and VR problems, and spent a significantly longer time lingering on MR [χ^2^ (2) = 21.6, *p* < 0.001] and VR problems [χ^2^ (2) = 21.6, *p* < 0.001]. Collectively, these preliminary results suggest that the reason the participants did not seek help was not the absence of a need for help. Rather, the failure to seek help may imply a breakdown between metacognitive monitoring and control processes, namely that participants failed to execute the appropriate control strategy (i.e., help-seeking). For the participants who sought help, the time it took for them to eventually seek help could reflect the level of difficulty and the degree to which participants felt reluctant to do so.

We first conducted the reluctance analyses in a way that could reflect real-world help-seeking. In the real world, where help-seeking is not restrained or pressured by the experimental setting, the time to seek help is unlimited. Thus, participants’ decisions not to ask for help in the current study, which can reflect the highest level of difficulty in seeking help on one’s own, may change given the abundance of time and social resources in real life. Thus, to reflect the level of difficulty as intended to be captured with the reluctance measure, for those participants who did not seek help, we used the maximum time observed in the current study for each domain as the measure of their reluctance to seek help. Following this logic, the current analyses included all the participants (*n* = 143). The maximum time (MR Max = 1225.5 s; VR Max = 475.9 s) was used as the reluctance time it would take for participants who did not seek help in the study (63 and 52 participants for VR and MR, respectively) to hesitate before receiving any eventual help in the real world. For those who eventually ended up seeking help, we measured reluctance as the total time it took for them to do so. We then calculated the connection between help-seeking reluctance, domain, and impostor scores. Regression results suggested that, of marginal statistical significance, participants reporting higher IP scores took longer to seek help than those reporting lower IP scores on MR problems (r = 0.009, *p* = 0.07), but not on VR problems (r = −0.002, *p* = 0.90). Again, from an additional perspective, following the median split, the Kruskal–Wallis test results showed that High(er) Impostors held out for a longer time before seeking help on MR problems [χ^2^ (2) = 5.11, *p* = 0.08], but not on VR problems [χ^2^ (2) = 1.43, *p* = 0.49].

Another way to analyze the data is to look only at the participants who did ask for help with their reluctance measure. While this may be a more accurate measure to represent reluctance to seek help, we also acknowledge that a significant portion of the participants are lost (44.1% and 36.4% participants for VR and MR, respectively). Still, regression results suggested similar numerical patterns. Participants reporting higher IP scores took (numerically) longer to seek help than those reporting lower IP scores on MR problems (r = 0.015, *p* = 0.14). On the other hand, the opposite direction of association was reported for VR problems—participants with higher IP scores reported a numerically shorter time to seek help than those with lower IP scores (r = −0.023, *p* = 0.14).

While not definitive from the current data, we believe that these data point to a glimmer of a pattern that might be in line with what we had expected—the High(er) Impostors spent slightly more time before seeking help than the Low(er) Impostors, but only for the MR problems, which were more difficult and may be related to math anxiety, especially for the current sample of mostly women, as will be further discussed in the Section 4.

### 3.4. Does Help-Seeking Lead to Increased Performance?

Finally, we were interested to see if seeking help was beneficial. After all, if help-seeking is a type of metacognitive control, then it follows that appropriate help-seeking should be associated with a higher level of performance. A Pearson’s correlation test suggested that seeking help was, indeed, positively correlated with accuracy scores (r = 0.48, *p* < 0.001). Additionally, as illustrated in Figure 4, following the median split of IP scores, both High(er) Impostors (r = 0.47, *p* < 0.001) and Low(er) Impostors (r = 0.50, *p* < 0.001) reported similar benefits of help-seeking. This is a crucial piece of the current data, given that a lack of help-seeking might obstruct one’s potential for improved performance, regardless of individuals’ IP scores.

## 4. Discussion

### 4.1. Summary of Findings

First and foremost, the IP score distribution of the participants in the current study was higher than what might be expected on average. Our second finding suggested that domain was a significant factor when looking at the probabilities of seeking help—with the math domain resulting in more help-seeking than the verbal domain. However, we found hints of evidence—in numerical patterns—of impostorism levels interacting with help-seeking behavior, namely that Higher Impostors were more hesitant and less likely to seek help than Lower Impostors. We expand on these findings below.

According to Clance’s breakdown of IP scores, a median score typically occurs around a score of 60 [25]. Here, while a normal distribution was found, the median score was 66. Thus, the current findings may not be representative of a larger, more general population. The observed higher distribution of IP in the current study was consistent with findings addressing IP prevalence across genders. Despite mixed results on gender and IP prevalence [54], a recent meta-analysis showed that women scored more highly than men on impostorism, especially in Europe and North America, where the current study took place [29].

The discrepancy in the level of IP between the current data and previous data could be due to several factors, and we remark below on the role that gender might have played. We also note that while we label our IP groups as High(er) and Low(er) Impostors, we acknowledge that a more accurate categorization might be High and Moderate Impostors. Nevertheless, even within the current group, our goal—to understand help-seeking behavior—was met with somewhat complex results. In particular, we found a hint of evidence that impostor levels may negatively interact with help-seeking behavior, via reluctance in help-seeking, but only in the domain of MR. We expand on the discussion below.

Given that a significant portion of one’s learning takes place in a social environment—school—it is important that students learn how to seek help when they know that they do not know. In other words, a skilled metacognitive control strategy is vital for all learners. In the current research, we found that, indeed, when people ask for help, performance is boosted. On the other hand, the data showed that there are subtle differences when it comes to who will ask for help and when that help will be sought. On a positive note, the results showed that as the MR problems were more difficult than the VR problems, participants sought help appropriately—they knew that there was a gap in their MR-solving ability and, thus, sought more help. Perhaps more negatively, while not reaching statistical significance, we found a numerical pattern that indicated the possibility that High Impostors could be less likely to seek help on math problems when compared with Low Impostors. Indeed, the data showed that, compared to Low Impostors, High Impostors took more time to seek help. We believe that, unfortunately, the lack of strong statistical effects could be due to the high distribution in the IP scores of the current sample, which in itself is an important finding.

Our data were in line with the notion of math anxiety and followed past studies showing that those scoring more highly on math anxiety reported lower metacognition [50,55,56]. Adding to previous work on metacognitive failures in the math domain, High Impostors in the current study demonstrated a metacognitive breakdown between intact monitoring and failed appropriate control by being numerically less likely and more reluctant to ask for help with MR problems when needed, which consequently negatively affected their performance. Our findings, thus, highlight the potential interplay among math anxiety, impostor feelings, and the deployment of appropriate metacognitive control strategies in math problem-solving. Interestingly, there was no difference between the High Impostors and Low Impostors for help-seeking on VR problems. One possible explanation could be that, for VR problems, which were not thought to be very challenging, all participants felt “safe” seeking help. In other words, the notion of “hiding” one’s incompetencies may be a key factor in help-seeking. Thus, one of the main reasons for lower performance in certain domains may simply be because “not knowing” may be more shameful. To better understand the nuances of help-seeking in various domains—and whether some domains might bring about emotions such as shame—is unknown, and we look forward to future research targeting these issues.

This domain-specific anxiety may also manifest particularly for women, exacerbated in the context of a stereotype threat associated with MR problem-solving. Math stereotype threat, found to be especially prevalent among women, refers to the apprehension that their performance in math-related tasks might confirm negative stereotypes about their gender group [57,58]. Previous research has demonstrated the detrimental effects of stereotype threat on women’s performance and interest in math and other STEM subjects [59]. Consistent with these findings, the current study, conducted on a mostly female sample, with women students primarily enrolled in an all-women’s college, further gives rise to the domain characteristics of behavioral approaches to math learning. In addition to the impact on math performance and motivation, our results suggest that help-seeking behaviors may also be influenced by stereotype threats specific to women and in the math domain. Female students, concerned about being judged stereotypically in the math domain, might face greater challenges in seeking help, despite recognizing the need for external help. This domain characteristic, giving rise to the experience of IP among women in higher education, could be the main reason for their metacognitive breakdown not at the monitoring phase, but, rather, at the control phase.

Another explanation could be that STEM subjects are thought to be more individualist, discouraging communal behaviors such as collaboration and helping one another [60]. This is not surprising given that STEM instruction is often delivered in the form of class competition, where in order to succeed one has to outperform others on a grade curve. This emphasis on individualist goals and class competition, in turn, could foster impostor feelings. Moreover, compared to non-STEM subjects, STEM subjects have been more heavily associated with fixed, individual brilliance and intelligence [61]. Impostor syndrome, on the other hand, refers to the self-belief that one is as intelligent as one’s past accomplishments reflect [30]. Thus, the observed difficulties in help-seeking in our study suggest that the observed hesitancy and avoidance of asking for help with math could be due to the fear of being “found out” as less intelligent. 

Help-seeking can be a complicated issue. As was described in the introduction, certain groups of people may find it extremely difficult to ask for help, as it would indicate giving up or being lazy. However, we hope to highlight the importance of help-seeking strategies and their relation to difficult-to-measure factors, including impostorism and domain qualities. Both the learner and also educators need to understand when and why some students might never raise their hand or, if we were to consider a more general context, seek help during office hours [62]. Rather than writing it off as “checked out”, or lacking metacognitive awareness, there could be any number of factors that encourage them towards, or keep them from, exhibiting good control. High levels of anxiety and stereotype threat in the math domain, for instance, could potentially be a deciding factor for students’ academic decision-making [63].

### 4.2. Broader Implications

Overall, the studies give one pause for thought and allow us to wonder whether help-seeking barriers, however small, may be related to academic entitlement, which refers to the perception that an individual is entitled to more than one worked for, regardless of one’s effort or ability [64]. Low academic entitlement has been found to be related to a decreased level of help-seeking, as well as other demographic factors, acting as an intergenerational characteristic of the middle class [65]. Individuals from lower socioeconomic status groups, and women, consistently report lower levels of academic entitlement, which presumably is associated with the impostor feeling of being less deserving than others to reach for help when they need it [66,67]. Here again, we highlight the complexity of knowing what it means to succeed academically, and what we know about the relationship between help-seeking and performance, both issues that need to be better understood if they are to be addressed.

Ironically, a sense of entitlement has been given a bad reputation, where students who are seeking help may be described as reaping benefits without putting in their own efforts [68]. We believe the opposite—seeking help is not a “cover” for a lack of effort. On the contrary, entitlement is an expression of metacognitive control. It may be the impostors, not the entitled, who are trying to get away with a cover-up of sorts, i.e., the fact that they know that they do not know. While we do not present definitive data here, our results do hint at a diverging pattern. That is, there may be cases, even in scenarios as complex as learning, where seeking information when one is aware of a knowledge gap—the very definition of learning—may be restrained at times or in different domains. 

In such contexts, instead of critiquing academic entitlement, greater emphasis should be placed on intellectual or academic humility, a concept referring to the willingness to acknowledge one’s limitations and take action to consult others’ intellect [69]. The existing literature supports the positive role of intellectual humility in fostering mastery of learning behaviors, such as a higher likelihood of persisting after failures and an openness to others’ views [70]. In essence, this construct addresses both aspects of successful metacognitive learning: intact monitoring (i.e., being aware of one’s knowledge gap), as well as adaptive control strategies (i.e., persistent effort and valuing others’ intellect). Therefore, promoting optimal learning outcomes through the practice of intellectual humility may provide a solution to the challenges faced by students who experience impostor fears and domain-specific concerns, as observed in the current study with women students struggling to, or hesitating to, seek help for math reasoning problems.

### 4.3. Contribution

The present study investigated the challenges in help-seeking as a case in point of hindered metacognitive control, despite intact monitoring. The existing metacognitive literature primarily focused on academic help-seeking at the individual level, exploring factors such as academic entitlement, attributional style, and internal motivation [41,67,71]. These approaches often neglected the social dimension of learning, despite evidence suggesting that individuals’ monitoring of their social context can significantly impact their metacognitive control and subsequent learning behaviors [20]. Based on these findings, the present study focused on the social dimension of individuals’ metacognitive control strategies. Specifically, we investigated the role of the impostor phenomenon (IP)—a phenomenon that makes individuals feel like “frauds”, despite their objective achievements—in relation to students’ help-seeking behaviors across math and verbal domains. The numerical pattern in our results aligned with prior research indicating heightened vulnerability among High Impostors, particularly within the STEM field of math [10,72].

Our study also represents a pioneering effort to connect IP with potential metacognitive breakdown, namely obstacles to help-seeking behaviors. Furthermore, our findings on domain differences between MR and VR problem-solving emphasize the particular significance of addressing impostor feelings in STEM education.

### 4.4. Limitations

Despite its contributions to the novel link between IP and metacognition, the present study also faced several major limitations. One primary limitation was the lack of sample diversity, as participants were exclusively recruited from introductory psychology classes at a private all-women liberal arts college. Consequently, the sample may not be representative of the broader population. The observed patterns of help-seeking behaviors across math and verbal domains could be more reflective of highly educated young women, a group that has been underrepresented in, and discouraged from, the STEM field [30,36,39]. This homogeneous sample demographic could also account for the particularly high number of High Impostors in our study [30]—based on the original criteria for interpreting CIPS, more than half of the participants in the current study would, in fact, be classified as High Impostors [25]. Therefore, our findings essentially describe the difference in help-seeking behaviors between moderately and extremely high impostors. This lack of variation in IP scores could also be the reason for the limited interaction effect observed between IP and help-seeking in the current study, as well as exhibiting a glaring reason as to why research on impostorism is challenging.

In future research, it is crucial to incorporate a more diverse sample to address the potential significant limitations identified in the current study. Furthermore, it is important to recognize that help-seeking in the context of individual problem-solving represents only one aspect of the learning process, in which metacognition is practiced. To gain deeper and more thorough insights into the relationship between social context and individuals’ metacognitive learning strategies, future investigations may benefit from exploring alternative methodological approaches to measure students’ control behaviors. For example, involving a real-person presence in the experiment could provide a more comprehensive understanding of how IP relates to metacognitive processes in a social setting.

## 5. Conclusions

While there is no official understanding of “how and when to seek help”, the current research suggests that there may be differences in how people approach learning in social contexts, as well as differences in the domain of the content being learned. To maximize learning, help-seeking is a sure way to boost learning. Yet, the current study, as an early effort to bridge IP and metacognitive control, suggested that there may be differences in help-seeking decisions across impostor groups. We look forward to discovering more of the various factors that impact help-seeking. For instance, as women, East Asian students, and first-generation students are likely to score more highly on the impostor scale [47,73,74], a better understanding of the barriers to help-seeking, as well as the ways to knock those barriers down, is crucial. Overall, help-seeking should be thought of as an optimal metacognitive control strategy, one that each and every student has the potential, and the right, to express when needed.

## Figures and Tables

**Figure 1 behavsci-14-00810-f001:**
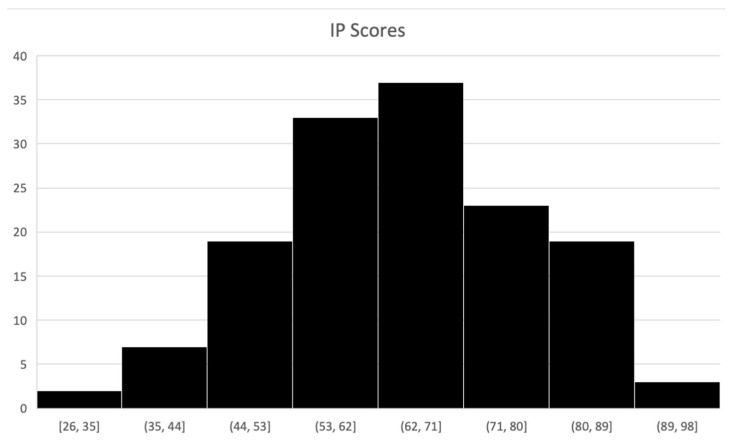
The frequency distribution of impostor scores across a continuum.

**Figure 2 behavsci-14-00810-f002:**
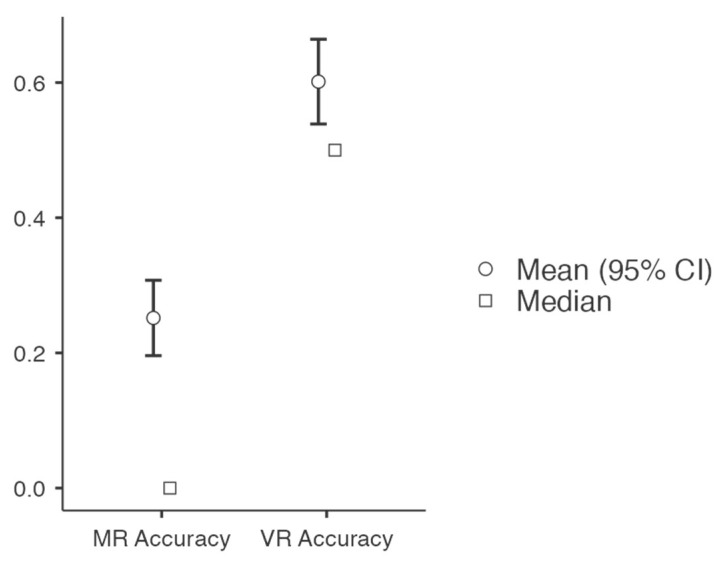
The distribution of accuracy scores across MR and VR problems.

**Figure 3 behavsci-14-00810-f003:**
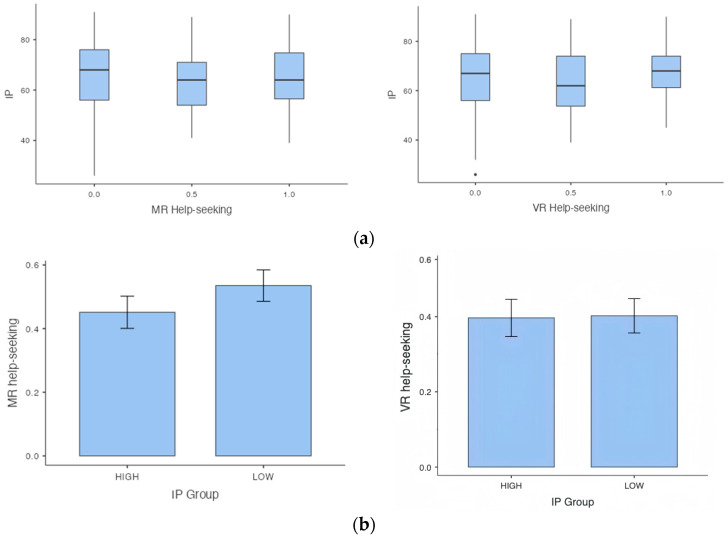
(**a**) The distribution of participants’ IP scores across different help-seeking probabilities for MR and VR problems. The help-seeking probabilities could be 0, 0.5, or 1, representing, respectively, help-seeking on zero, one, or two of the problems within a subject. (**b**) The distribution of help-seeking probabilities for high and low impostor groups across MR and VR problems.

**Figure 4 behavsci-14-00810-f004:**
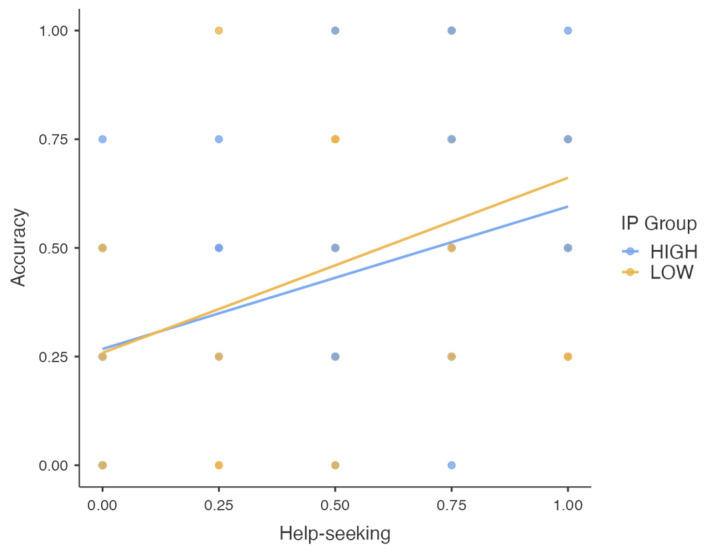
Accuracy scores as a function of participants’ help-seeking probabilities for Higher and Lower Impostors. Both the help-seeking probabilities and the accuracy scores had the values of 0, 0.25, 0.5, 0.75, or 1, representing, respectively, help-seeking or getting the correct answers on zero, one, two, three, or all four problems. The best-fit lines suggest a positive relationship between help-seeking probabilities and accuracy scores for both groups.

## Data Availability

If the manuscript is accepted, the data will be made publicly available at an appropriate outlet.

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
