# Peer review of "High Impostors Are More Hesitant to Ask for Help"

_behavsci, 2024, doi:10.3390/bs14090810_

Round 1

Reviewer 1 Report (New Reviewer)

Comments and Suggestions for Authors
  •  

Summary and general comments:

This is a very interesting study which makes - to the best of my knowledge - an original contribution to the field through exploring the relationship between IP and help-seeking behaviour. In a small number of places, discussion of key concepts relating to the focus of the study needs to be provided in greater depth to fully demonstrate how this study builds upon existing knowledge. For example, current knowledge regarding IP and maths is very brief during the opening sections of the paper, given its importance and relevance for this study. Further examples of places where more detail is needed are provided in the 'Specific comments' below.   

Specific comments:

P.4, line 161-163. The overview of literature relating to IP and maths is sparse here given the focus of the paper, and would benefit from further elucidation.

P.6, line 295.  Not all studies suggest higher rates of IP in women compared with men (for example, see the systematic review by Bravata et al., 2019), so it would be useful to strengthen the rationale for suggesting this as a possible explanation here. This could be done through further reference to the data, or to wider reading to support this assertion (or by signposting the later discussion at this point). 

P.7, line 365. The assertion relating to maths anxiety and women needs more support here to increase credibility. Alternatively, signposting could be added to show that further discussion will be provided later within the paper. 

It would also be important to review the section on p.3 (lines 117-119) as this appears to contradict your suggestions here with regard to the prevalence of IP in women, and therefore the influence of the larger proportion of women within your sample.

P.11, line 439. How does this compare with previous studies?

P.11, line 485. Please define 'academic entitlement' for those unfamiliar with this term. 

Comments on the Quality of English Language

This paper was clear and accurately written. 

Author Response

Summary and general comments:

This is a very interesting study which makes - to the best of my knowledge - an original contribution to the field through exploring the relationship between IP and help-seeking behaviour. In a small number of places, discussion of key concepts relating to the focus of the study needs to be provided in greater depth to fully demonstrate how this study builds upon existing knowledge. For example, current knowledge regarding IP and maths is very brief during the opening sections of the paper, given its importance and relevance for this study. Further examples of places where more detail is needed are provided in the 'Specific comments' below.   

Response: Thank you for your insightful comments and for recognizing the originality of the study. To address your suggestion for a more in-depth discussion of key concepts, we have focused on expanding the literature review and discussion on STEM anxiety, gender, and academic entitlement.

Specific comments:

P.4, line 161-163. The overview of literature relating to IP and maths is sparse here given the focus of the paper, and would benefit from further elucidation.

Response: Thank you. We have expanded on the literature on domain-specific IP findings (see lines 166 to 183), addressing the current study’s rationale for focusing on women and math.

P.6, line 295.  Not all studies suggest higher rates of IP in women compared with men (for example, see the systematic review by Bravata et al., 2019), so it would be useful to strengthen the rationale for suggesting this as a possible explanation here. This could be done through further reference to the data, or to wider reading to support this assertion (or by signposting the later discussion at this point). 

Response: Thank you for the comment. For the results section, we pointed out this higher distribution observed in the current sample of women in higher education was in line with the origin of the construct (see lines 317 to 320), the implication of which was later discussed in detail (see lines 470 to 473 and lines 516 to 531) in the discussion section.

P.7, line 365. The assertion relating to maths anxiety and women needs more support here to increase credibility. Alternatively, signposting could be added to show that further discussion will be provided later within the paper. 

Response: Thank you for the suggestion. In the result section, we added signposting on the finding that implies math anxiety (see lines 413 to 415). The discussion of this assertion can be found in lines 499 to 507 and lines 516 to 531.

It would also be important to review the section on p.3 (lines 117-119) as this appears to contradict your suggestions here with regard to the prevalence of IP in women, and therefore the influence of the larger proportion of women within your sample.

Response: We have now included more comprehensive evidence on the prevalence of IP across genders to present a more rounded literature review on the topic (see lines 119 to 126).

P.11, line 439. How does this compare with previous studies?

Response: Thank you for raising this question. We clarified the relationship of current findings to previous work (see lines 500 to 507). Specifically, we stated that the current findings expanded on the existing body of work on how math anxiety and IP may be the contributing factors for metacognitive failures in the control phase.  

P.11, line 485. Please define 'academic entitlement' for those unfamiliar with this term. 

Response: We have now added the definition of the term (see lines 557 to 559).

Reviewer 2 Report (New Reviewer)

Comments and Suggestions for Authors

I thoroughly enjoyed reading this article! Impostor Syndrome is an area of interest for me, and your study is a great contribution to this body of literature. I thought your design was meaningful and your results and discussion were well written. Thanks for your time spent researching this topic! 

Author Response

I thoroughly enjoyed reading this article! Impostor Syndrome is an area of interest for me, and your study is a great contribution to this body of literature. I thought your design was meaningful and your results and discussion were well written. Thanks for your time spent researching this topic! 

Response: Thank you for your encouraging feedback and the time you took to engage with our study. We truly appreciate your support in our work.

Round 2

Reviewer 1 Report (New Reviewer)

Comments and Suggestions for Authors

Thank you for your amendments to this submission - I very much enjoyed reading this.

This manuscript is a resubmission of an earlier submission. The following is a list of the peer review reports and author responses from that submission.

Round 1

Reviewer 1 Report

Comments and Suggestions for Authors

This is an interesting paper about the metacognitive control of students who exhibit imposterism, and especially the help-seeking behaviours of such students when addressing tasks in mathematics and verbal categories. The very useful findings highlight a difference between High and Low imposterism students on help-seeking in relation to mathematics problems but not in relation to verbal problems. The study is worthwhile and interesting, and I do believe that the audience can find an audience in areas of scholarship related to the Behavioral Sciences journal.

In my view, the paper is well-written overall, though certain aspects are underexplained. These aspects occur mostly at the start and end of the paper, with the central sections being much stronger in my view. I therefore think that the authors should be asked to address the following comments:

·         The Abstract could be strengthened by situating the paper in the literature on metacognition (at the beginning) and emphasising the contribution the paper makes to this literature (at the end).

·         The Introduction section doesn’t really say much about the paper but rather addresses the broad topic. I think it would be useful to say directly what the paper is about and what the aims of the paper are.

·         Sections 1.1 and 1.2 are diligent literature reviews. But I think that each section would benefit from comments (perhaps around lines 55-62 and 122-127) about (a) the strengths of existing work on which the present paper builds, and (b) the limitations of the present work, thereby setting up a potential contribution that this paper aims to make.

·         Somewhere in these literature review sections it would also be worth explicitly addressing the matter of whether the literatures on metacognition and the imposter phenomenon are usually seen as linked, or whether linking these two areas of literature is a novel ambition of this paper.

·         Early in section 1.3 a variety of hypotheses are set out (lines 132-134, and 153-156). It would be useful to have some *explicit* text on how these hypotheses were derived from the literature reviews reported in the earlier sections.

·         It would be useful also to justify why the paper focusses on *mathematics* problems. Ideally, this would have been highlighted as an issue in section 1.1 and/or 1.2, but this has not been done in the present version.

·         It would be useful in this section to justify the choice of the CIPS scale in more detail. Why was this scale chosen? To what extent does this study share similarities with, and differ from, previous studies that have made use of this scale?

·         In section 2.1, when reflecting on the participants, it would be useful to reflect on the potential implications that the students are all female and that they all study Introductory Psychology at a liberal arts college in the USA. Might this have implications for a study on help-seeking, and differences between mathematics and verbal problems? Reflect here and also at the end under limitations as appropriate.

·         In section 2.3, it would be useful to reflect on the design of the various problems the students were presented with. Perhaps these problems could be briefly documented too, possibly in a table?

·         In section 2.4 the notion of “asking for help” is discussed. However, surprisingly, the reader cannot know what help was actually provided when students did ask for help. This would surely have in impact on help-seeking behaviours. Therefore, this should be justified and documented.

·         Section 3 would benefit from some additional signposting. What will be presented, in what order, and how does this order allow the authors to address their research priorities?

·         The section numbering in section 3 seems inconsistent. The sections should be re-numbered and the figures distributed into the text as appropriate.

·         Section 4 requires, in my view, an extra subsection, dealing with contributions to literature. What distinctive new contributions does this paper make to the two areas reviewed earlier (in sections 1.1 and 1.2), and potentially to drawing out links between these areas?

·         Section 5 should discuss limitations (including the potential limitation discussed above about the participants) and briefly sketch a plausible trajectory for future work.

Reviewer 2 Report

Comments and Suggestions for Authors

The manuscript “High Imposters are more hesitant to ask for help” focuses on an interesting topic with relevance in the educational realm. The importance of the social context of learning and on the use of metacognitive strategies is a research topic to be followed.

The overall manuscript is well-written and follows a coherent structure of presentation. However, some aspects could be further explained to improve the clarity of the arguments presented. Please see the following suggestions:

1.      It is unclear in point 1.1 why the experience/scenarios are starting to be introduced. It is confusing and does not help to support the argument that was to be made at that point. It would be relevant to extend the argument about the help-seeking in learning because it would be the base for the research questions, and it is sparse.

2.      In the next subsection, it would also be helpful to start revising the construct of the Impostor Phenomenon by exploring some definition and if wanted to make the comparison with the Impostor Syndrome known in lay literature to add a clearer distinguishing for both concepts.

3.      The relevance of the study that was conducted should be better stressed. It is not clear while reading the manuscript the gap in the literature that are be addressed by the study presented and why is this contributions is relevant for the field of study.

4.      How were the participants recruited? How old were the participants? It is implicit that the participants are women, but it would be relevant to be more explicit about the characteristics of the participants who were recruited for the experience.

5.      Regarding the ethical procedures, the authors could be more explicit about the reward given to the participant and the ethical assurances given before, during and after the proceedings.

6.       A section about analytical procedures would help to understand the statistical options assumed in the results section. For instance, why were used exclusively non-parametric tests?

The data seems to allow you to cross the variable of help-seeking behavior and the one for groups of Impostor Phenomenon to give a more comprehensive notion about the combination of the two variables that are the base of the research questions. The analysis as they are presented seems to lack support for the conclusions that are to be made.   
